# Determining resources and capabilities in complex context: A decision-making model for banks

**Mochammad Ridwan Ristyawan**[1,2*], **Utomo Sarjono Putro**[1], **Manahan Siallagan**[1]

**1** School of Business and Management, Institut Teknologi Bandung, Bandung, Indonesia, **2** Faculty of Economic and Management, Universitas Tanjungpura, Pontianak, Indonesia

* mochammad_ridwan@sbm-itb.ac.id and m.ridwanristyawan@untan.ac.id

## Abstract

The role of resources and capabilities in shaping and implementing a firm's strategy is paramount. The COVID-19 pandemic underscored the necessity for managers to possess a decision-making model that facilitates the selection of resources and capabilities in a real-time, dynamic, adaptive, and iterative manner. However, the dynamic capabilities framework, which serves as a decision-making model, faces three significant issues when selecting resources and capabilities within complex contexts. These issues, identified as research gaps, include context mismatch, inappropriate treatment, and strategy alignment. These gaps serve as the foundation for decision making models. This study aims to develop a decision-making model for determining banking resources and capabilities. The novelty of this study is encapsulated in the proposed decision-making model for resource and capability determination in complex contexts. Furthermore, this study employed a methodology adapted from the International Society of Pharmacoeconomics and Outcomes Research—Society of Medical Decision Making (ISPOR-SMDM). The research methodology was conducted in ten stages to develop a decision-making model. This study used qualitative methods, a case study strategy, and an abductive approach. The research sample consists of Indonesian State-Owned Banks (SOB). This research culminated in a proposed decision-making model that includes seven managerial decisions: probe, sense, structuring, bundling, building, leverage, and reconfiguring. This model integrates fuzzy preference judgments as inputs, deep learning analytics (predictive analysis) as processes, and success rate predictions as outputs. Theoretically, this research contributes to the enhancement of dynamic capabilities through the complex domains of the cynefin framework. Practically, it offers a decision-making model for the board of directors (BOD) to determine resources and capabilities amid complex environmental changes.

**Data availability statement:** All relevant data are within the manuscript and its Supporting Information files.

**Funding:** The author(s) received no specific funding for this work.

**Competing interests:** The authors have declared that no competing interests exist.

## Introduction

Resources and capabilities play a significant role in determining and executing a firm's strategy [1]. According to Barney's resource-based view (RBV), decision-making is integral to the preparation and utilization of resources to align with strategic objectives [2]. Specifically, the decision-making process within the RBV, referred to as the dynamic capabilities concept by Teece et al. [3], must account for the decision-making context in a complex environment [4]. However, Teece's framework is inadequate for detailing the decision-making process concerning the volatility, uncertainty, complexity, and ambiguity (VUCA) of the environment [5,6]. Moreover, managers require a decision-making model that incorporates data technology to facilitate decision making regarding resources and capabilities under complex conditions [7].

Based on the cynefin framework by Snowden and Booone, decision makers are advised to identify the environmental context to formulate an appropriate decision-making model [4]. The cynefin framework delineates four distinct contexts within the decision-making paradigm to facilitate understanding of problem situations. These contexts are categorized as simple, complicated, complex, or chaotic. It is imperative for decision-makers to ascertain the specific context of decision-making in relation to environmental conditions to ensure optimal decisions, particularly concerning resource and capability allocation.

Previous studies have provided frameworks and models to determine resources and capabilities in complicated contexts [1,5–7]. The logical structure of the dynamic capabilities framework, a sample of the decision-making model for resources and capabilities by [5], is the most cited reference for decision-making to form resources and capability configurations [6]. This model is inadequate for accommodating the complex context of decision making, where the dynamic environment requires a real-time, dynamic, adaptive, iterative model [6]. The logical structure of the dynamic capabilities framework proposed by Teece et al. [5] illustrates the managerial decisions (*sense*. *seize* and *transform*) in a complex context adopted from the micro-foundation of the dynamic capabilities concept [5,8,9]. However, this model is not conformable with the complex context of the cynefin framework, which starts by *probing* the factors, *sensing* the pattern of cause-effect factors, and *responding* with actions. [4]. These disputes in the decision-making context have caused this model to not run well under complex conditions, such as during and after the COVID-19 pandemic.

For example, during and after the COVID-19 pandemic, directors and managers recognize the inadequacy of traditional decision-making methods, which typically involve annual meetings to identify market opportunities, capitalize on them, reallocate resources, and implement strategies [5,8,9]. In various instances, numerous airlines have been compelled to reduce flight routes and operational hours owing to restrictions on human mobility [10,11]. Because of the public's reluctance to travel during the pandemic, AirAsia experienced significant revenue losses [11,12]. In response, the CEO shifted the company's strategic focus towards the food industry to ensure its sustainability. This strategic pivot was perceived as a reactive measure for the AirAsia Airlines. It is imperative that AirAsia's strategic decisions are informed

by resource data, enabling the rapid formulation of new strategies through computational analysis [10,12]. The COVID-19 pandemic underscored the necessity for managers to be supported by an appropriate decision-making framework to optimize resource and capability configurations under VUCA conditions.

The banking industry faced the same situation: bank debtors faced bankruptcy and disrupted bank operations during COVID-19 [13]. Hence, banks should adjust their decision-making processes [9,13]. A sample case of SOB in Indonesia revised their strategy during the COVID-19 pandemic through an annual work meeting involving all managers [14]. This decision-making process is time consuming, and decisions must be formulated immediately. A quick resource decision-making model is required by a company to adaptively and efficiently provide resource decisions [10].

Fig 1 presents a comparative analysis of the decision-making processes in complicated versus complex situations within the dynamic capabilities framework. Notable distinctions exist between the two contexts. The COVID-19 pandemic exemplifies a complex environmental context, whereas its absence characterizes a complicated context. As noted by [4,9], managerial decisions within the dynamic capabilities framework resemble those in complicated contexts, following a sequence of "sense," "analyze," and "respond." According to [4], Fig 1 illustrates that the BOD can discern cause-effect relationships in resource selection within a complicated context, ensuring alignment with the bank's strategic objectives. Furthermore, the BOD is capable of identifying ordinary resources and capabilities as well as valuable, rare, inimitable, and non-substitutable (VRIN) resources and dynamic capabilities [5,10,15]. This identification is facilitated by their knowledge and experience regarding the cause-effect patterns of strategy and resources/capabilities under normal

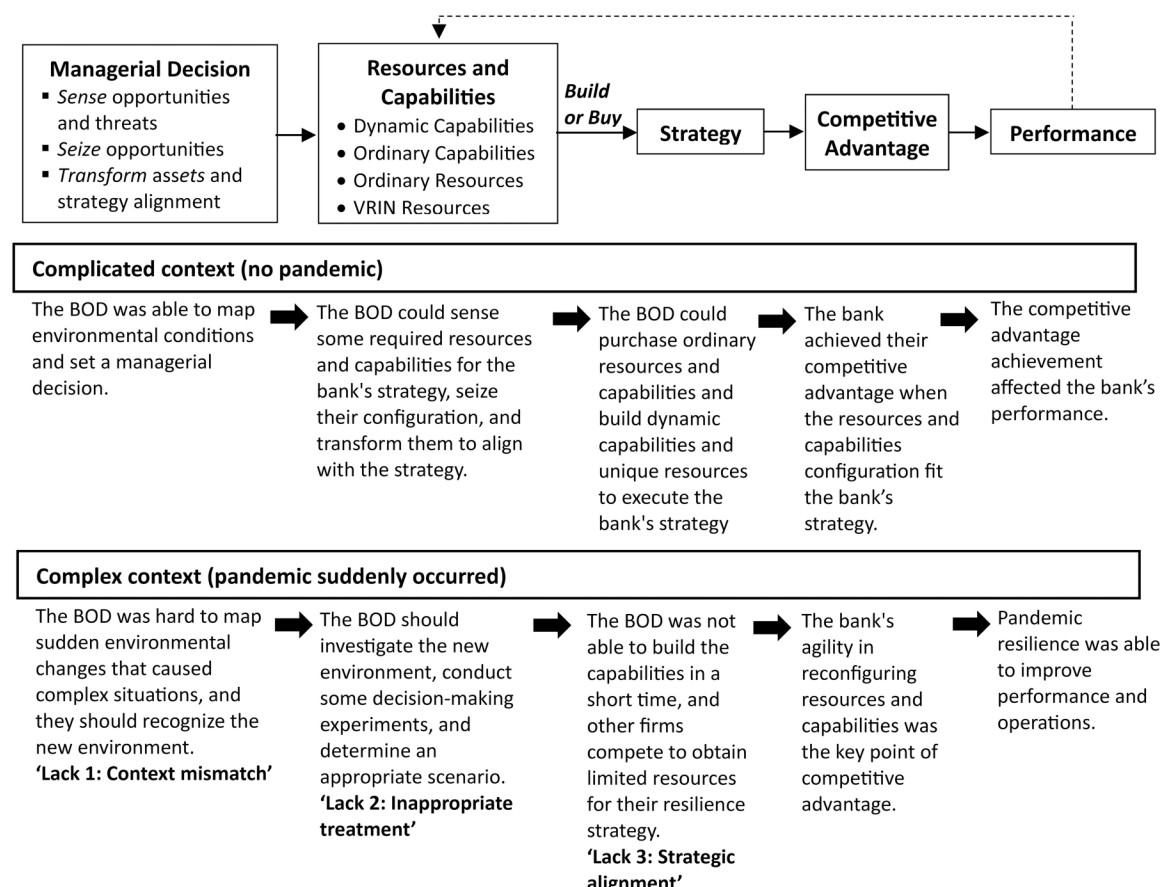

**Fig 1. The Comparison between complicated and complex context in dynamic capabilities framework.** Source: [5] (Simplified Framework).

conditions. Consequently, decision-making processes can incorporate both "build" and "buy" strategies in regular resource decision-making. Ultimately, the alignment between resources and strategy enhances a bank's competitiveness and performance.

However, the BOD has not yet formulated a strategy that effectively navigates rapid environmental changes, as outlined in the dynamic capabilities framework. It is imperative for BOD to examine the relationship between resources, capabilities, and strategies that drive successful organizational performance [13,14]. The decision-making approach within the dynamic capabilities framework, which encompasses sensing, seizing, and transforming, has been inadequate for addressing the complexities and volatility of the current market environment [4,16,17]. As noted in [4], intricate circumstances necessitate that the BOD explore the cause-and-effect relationships between resource configuration and strategic planning. The BOD must discern the optimal combination of resources and capabilities, and adjust to environmental shifts by reconfiguring these elements to implement the strategy. The primary issue is *context mismatch* in decision-making [4,14]. The second issue pertains to *inappropriate treatment*, where managerial decisions must be tailored to the complex environmental context, necessitating thorough information gathering for informed decision-making [14]. The third issue underscores the importance of *strategic alignment*, emphasizing that any strategic shift must be accompanied by a bank's flexibility in determining resources and capabilities [2,9,10,13,18].

Based on previous insights, gaps exist in *context mismatch*, *inappropriate treatment*, and *strategy alignment*. The framework outlined in [5] requires modifications to effectively address the issues identified in [14]. The existing framework lacks the capacity to adequately elaborate on these three issues within dynamic environmental contexts and human behavior interactions. Consequently, the development of a new conceptual framework and proposed model should incorporate the decision-making complexities inherent in the cynefin framework alongside technological applications that facilitate real-time, dynamic, adaptive, and iterative decision-making processes [6,7,19,20]. Based on these insights, the following question arises: *"How can a decision-making model be developed to determine resource and capability configuration within a complex context?"* represented the first research question (RQ1).

Furthermore, a literature review [6] offers insights into decision-making mechanisms for determining resources and capabilities within a complex context. Ristyawan et al. identified several key factors for future research in the development of decision-making models to ascertain resources and capabilities [6]. Consequently, this study referenced Ristyawan et al.'s work by employing key factors to construct a decision-making model. The researchers formulated the following research question: *"What are the key factors to develop the decision-making model?"* for the second research question (RQ2).

To investigate these key factors as a foundation for model development, interconnected analysis is required. The research question formulated to guide this analysis is: *"How can key factors be interrelated to develop a conceptual framework?"* represented the third research question (RQ3). The analysis of interconnected key factors elucidates the relationship between the decision-making processes and technology. Consequently, this third question suggests that the interconnected analysis requires the application of Socio-Technical System (STS) theory to adapt the key factors identified in the study by Ristyawan et al. [6,19,21].

In the development of a conceptual framework, it is essential to integrate insights from both literature and empirical studies [22,23]. The literature review aids in identifying key factors, while empirical research serves to validate these factors with evidence from various sources, thereby constructing a conceptual framework for decision making in determining resources and capabilities within complex contexts. Consequently, a research question, designated as the fourth research question (RQ4), was formulated by the researchers: *"What is the conceptual framework of the decision-making for determining resources and capabilities?"* This question specifically addresses the conceptual framework for decision making. The development of this framework takes into account the technology employed in decision-making, drawing on the cynefin framework and the STS theory [4,19,21].

Furthermore, the cynefin framework facilitates the development of elements within the decision-making model that are derived from the conceptual framework [4,7]. This model development aligns with the complex

decision-making context inherent in the cynefin framework, characterized by the pattern of "probe"-"sense"-"respond" [4]. Therefore, the researchers formulated the final question as follows: *"How is the proposed decision-making model for determining resources and capabilities?"* to represent the fourth research question (RQ5). These five research questions aim to develop a decision-making model that assists managers in allocating their resources and capabilities to implement new strategies, thereby creating a competitive advantage and enhancing bank performance, particularly in complex contexts. Therefore, resources and capabilities must be meticulously prepared for a successful strategy execution.

The objective of this study is to develop a decision-making model for banks that determines resource and capability configurations within a complex context. The urgency of this research arises from the necessity for managers to have a rapid decision-making process to ascertain their resources and capabilities under dynamic environmental conditions [6,10,16]. Managers require a reference decision-making model when confronted with volatile environmental conditions. The proposed model serves as a prototype of a decision-support system that aids managers in determining resources and capabilities in a dynamic environment. A decision-making model should also consider context, human behavior, social interactions, decision characteristics, outcomes, and decision-support tools [6,17,21,24].

This study underscores the novelty of a decision-making model designed for banks to ascertain optimal configurations of resources and capabilities. This model facilitates decision-making in real time and is characterized by adaptability, dynamism, and iterative processes. While numerous scholars have endeavored to address these issues, the pandemic has altered human behavior, rendering existing decision-making models ineffective in determining resources and capabilities. Consequently, this study aimed to develop a decision-making model capable of orchestrating resources and capabilities through a participatory decision-making process in complex scenarios. Many scholars have focused on decision-making concerning resources and capabilities within complicated contexts, where information and knowledge are readily available to formulate various resource and capability preferences.

## Related works

### The cynefin framework

Managers must comprehend the management approach pertinent to decision-making strategies in complex environments [17,25]. The cynefin framework underscores the significance of the decision-making context when leaders are tasked with identifying the optimal resolution for a given problem [4]. This framework elucidates the cause-effect relationships inherent in issues, thereby aiding decision makers in formulating their managerial actions [4]. According to the cynefin framework, the decision-making context is categorized into five domains: simple, complicated, complex, chaotic, and disorder [4,25]. Four of these domains—simple, complicated, complex, and chaotic—require decision makers to thoroughly investigate situations and implement suitable contextual actions [4,17,25]. Conversely, the domain of disorder is applied when the cause-effect relationships of the problems remain ambiguous [17].

Furthermore, the cynefin framework delineates managerial actions pertinent to the four domains characterized by cause-effect relationships [17]. The 'simple' domain represents the realm of "known knowns," wherein decisions are unchallenged due to a shared understanding among all parties [4,17]. Managerial actions within this simple domain involve sensing, categorizing, and responding. The 'complicated' domain corresponds to the realm of "known unknowns" [4]. This domain requires decision makers to sense, analyze, and respond to the issues presented [4,17]. Subsequently, the 'complex' domain embodies the realm of "unknown unknowns," which has significantly influenced contemporary business practices [4]. In this domain, decision makers are required to probe, sense, and respond to emerging issues [4,17]. Finally, the 'chaotic' domain denotes the realm of "unknowables," where cause-effect relationships are indeterminable [4]. Here, decision makers must act, sense, and respond to the challenges encountered [4,17]. The cynefin framework [4,17,25] has been employed to refine the logical structure of Teece's dynamic capabilities framework [5], thereby aligning it with the complex domain of decision-making contexts.

## RBT and dynamic capabilities framework

Resource-Based Theory (RBT) posits a framework for understanding how a firm's resources impact its performance. This theory underpins Teece's development of the dynamic capabilities concept and provides a framework for dynamic capabilities [3,5]. Initially known as the RBV, RBT was pioneered by [26], with subsequent extensions by [2,27–29], building on Penrose's work. RBT has evolved from focusing on how resources enable a firm to gain a competitive advantage to examining how a firm can organize its resources and capabilities to enhance its performance [30]. Barney, a notable scholar in RBT, introduced the RBV tenets of unique resources: valuable, rare, inimitable, and non-substitutable (VRIN) [2]. [31] clearly differentiates between resources and capabilities within a firm and the associated RBV in behavioral decision theory.

[5] developed a dynamic capabilities framework that guides the strategic allocation of resources and capabilities. [32] advanced dynamic capabilities by incorporating micro-foundations that highlight decision-making processes to determine resources and capabilities. Dynamic capabilities refer to a firm's ability to integrate, build, and reconfigure internal competencies, including resources and capabilities, to achieve competitive advantage in a dynamic business environment [3,9,10]. Within RBT, dynamic capabilities represent a decision-making concept where managers endeavor to sense, seize, and transform resource and capability configurations [9]. Managers orchestrate the configuration of resources and capabilities to identify the optimal combination for executing a firm's strategy.

Other scholars, such as Sirmon et al., have sought to develop resource orchestration, extending the work on dynamic capabilities [8,33]. They formulated a decision-making framework for resource and capability determination, emphasizing resource management for value creation [8]. They also underscore that firms can attain competitive advantage by successfully establishing value creation. [8] introduced a resource orchestration framework that elucidates how a firm organizes assets and combines resources, detailing how managers orchestrate resources to secure a competitive advantage.

## Decision-making mechanisms and socio-technical system theory

A decision-making model must elucidate the mechanisms involved in selecting resources and capabilities [6]. It is imperative to clarify these mechanisms to guide the determination and optimization of resource configurations. Ristyawan et al. emphasized resource decision mechanisms, where the focal perspectives of resource decision making converge into context, processing strategy, mechanism characteristics, subject field, industry field, and decision support tools [6]. Their study referenced the cynefin framework to highlight the decision-making mechanisms pertinent to resource and capability determination in various fields [4,6].

Concurrently, a decision-making model should employ an STS approach to elaborate on the integration of technology and human behavior in the decision-making process, thereby elucidating the bounded rationality of decision makers in accessing problem data and information [19,34–36]. STS represents an organizational theory that explores the interrelated functioning of an organization's social and technological subsystems [19]. These systems delineate an organizational system into six dimensions: 1) people, 2) infrastructure, 3) technology, 4) culture, 5) processes/procedures, and 6) goals [21]. STS theory aids designers in determining subsystem structures and their potential roles as well as in understanding how new technology may be integrated with existing and planned social systems [19,34].

The STS approach can enhance the decision-making mechanism by designing an interrelated model of decision maker behavior and IT, providing a real-time, adaptive, dynamic, iterative approach [21]. This model addresses the acceleration of resource decisions for companies operating in unstable and complex environments [19]. This method serves as a bridge between the decision-making systems and computer technology.

## Bounded rationality in decision-making process

The concept of bounded rationality is central to the decision-making process [37]. Bounded rationality posits that a leader operates with incomplete information regarding alternatives, navigates complexity and uncertainty, and possesses limited knowledge and capacity concerning the consequences of actions [35,37–39]. According to [40], it is essential for a

leader to rely on subordinates to gather the information necessary for decision-making. Leaders can assess alternative decisions through the value judgments of their subordinates [35,37]. The assertion by [40] underscores the presence of bounded rationality in decision-making processes, particularly when decision makers confront environmental complexity and uncertainty.

Moreover, a heuristic decision-making model can be employed to address bounded rationality, wherein company leaders or boards of directors engage experts such as head divisions and managers to evaluate numerous alternatives [37,41,42]. While tangible resources can be physically quantified by managers, intangible resources and capabilities present measurement [3,31,43]. Consequently, decision making within the RBT framework adopts Simon's heuristic model, incorporating the expert value judgments of Tversky and Kahneman [37,41,42].

### Fuzzy preference relations in heuristic decision-making model

Fuzzy preference relations serve as a method for evaluating value judgments in the decision-making process [44–46]. This process encompasses social interactions among decision makers, each possessing varying levels of knowledge regarding decision alternatives, to arrive at an optimal decision [37,47]. Knowledge of these alternatives is encapsulated in the preference judgments of different choices [44,45]. The connection between social systems and technical analysis of decision-making is facilitated by the preference relations of certain alternatives [21,34,45]. These preference relations form the nexus between value judgments among decision makers within a heuristic decision-making model [37,41,42,46].

Fuzzy preferences represent the degree of preference that decision makers assign to alternative decisions [44,45]. Therefore, fuzzy preference relations act as intermediaries between preference degrees in a heuristic decision-making model [37,41,42,45]. This heuristic model is particularly suitable for decision-making contexts, where the BOD endeavors to gather comprehensive information and knowledge to navigate a complex environment [4,35,46]. Additionally, the fuzzy preference relation model can facilitate decision making in dynamic, real-time, adaptive, and iterative contexts [6].

Furthermore, fuzzy preference relations can aid in the formulation of resource and capability configurations that align with strategic objectives and foster a competitive advantage [5,44,45]. An effective resource configuration is derived from the optimal fuzzy preference relationships of resources [45,48]. The optimal outcome of resource configuration reflects a consensus among decision makers based on their preferences [37,38,41,44]. Thus, the fuzzy preference relations method seeks to achieve unanimity among managers to reach a consensus on resource configurations [44,45,48].

## Materials and methods

This study utilized qualitative methodologies to construct a decision-making model [47,48]. The abductive approach facilitated the analysis of qualitative data [47]. The development of this decision-making model was applied to a case study of a banking institution serving as a research strategy [50]. An SOB was selected as the sample for this study, focusing on the decision-making challenges encountered by banks regarding resource selection and strategy revision during the COVID-19 pandemic [50,51]. Additionally, a decision-making model was formulated through qualitative analysis, incorporating both primary and secondary data [50]. Primary data were gathered via in-depth interviews conducted within a focus group discussion (FGD) with participants selected through a purposive sampling technique [49,51,52]. The participants were five experts in decision-making, banking, and strategic planning [50]. Secondary data were obtained from a literature review that examined decision-making processes to identify resources and capabilities [49,50]. A literature search was conducted using Scopus, a reputable journal database [53]. Content analysis was employed to identify the key decision-making factors within the selected RBV literature [49,50]. This study adhered to the methodology outlined by the International Society of Pharmacoeconomics and Outcomes Research Society of Medical Decision Making (ISPOR-SMDM) to develop a conceptual model [23]. Fig 2 illustrates the research methodology employed to develop the resource decision making model. A diagram of the methodology is utilized to construct the conceptual framework, and the proposed model is presented as follows:

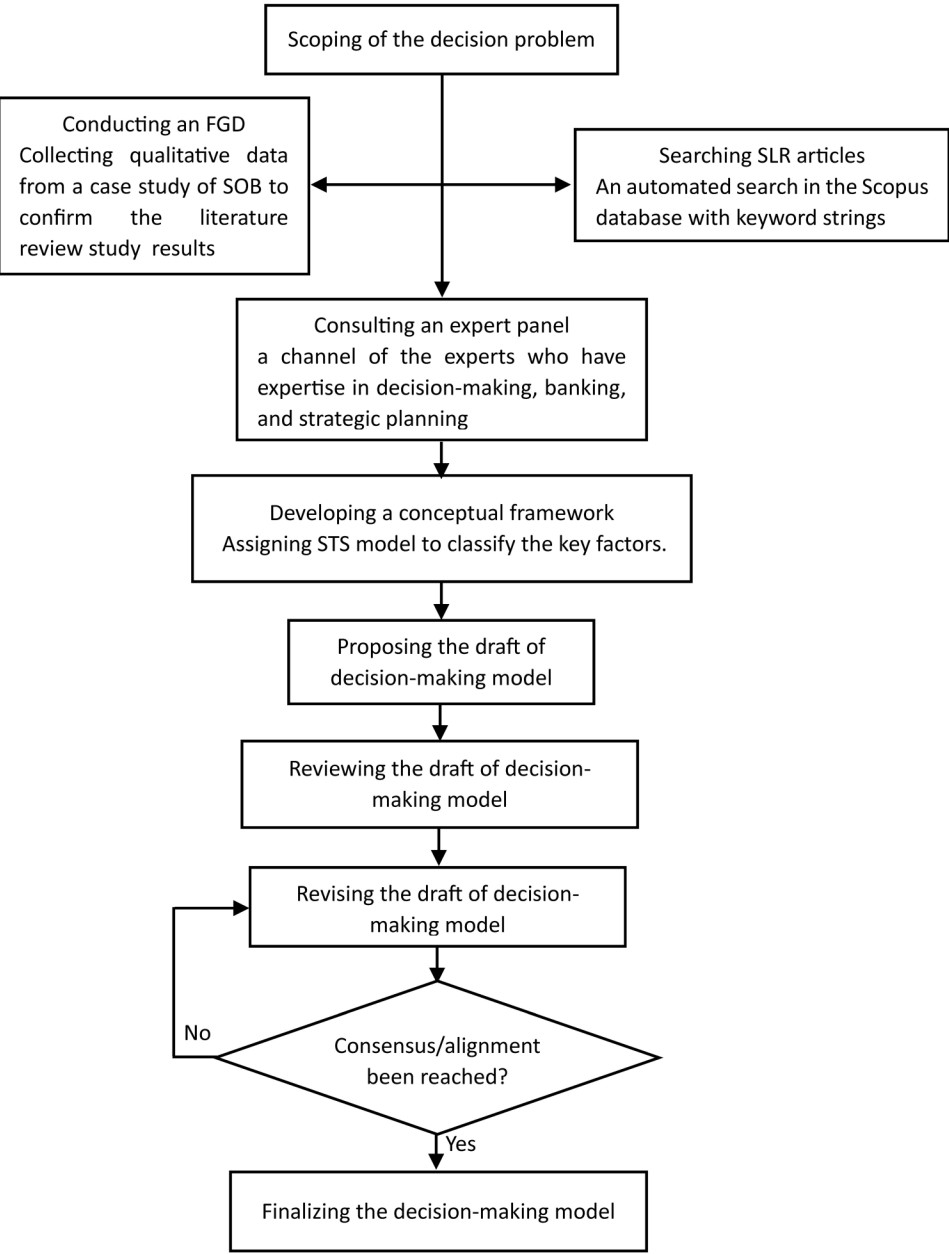

**Fig 2. Research Methodology of Conceptual Model Development.**

Based on the ISPOR-SMDM, this study developed a methodology comprising ten stages: scoping the decision problem, searching for systematic literature review (SLR) articles, conducting an FGD, consulting an expert panel, developing a conceptual framework, proposing a draft of the decision-making model, reviewing the draft, revising the draft, reaching consensus, and finalizing the decision-making model [23]. The research plan was executed over a six-month period. Researchers allocated one month to scoping the decision problem, one month to search for SLR articles, one month to conduct an FGD, two months to consulting an expert panel, formulating principles for decision-making model development, and proposing a draft, and one month to review, revise, align, and finalize the decision-making model. This

methodology was designed to address RQ1, which pertains to the development of a decision-making model that determines the resource and capability configurations in a complex context.

### Scoping of the decision problem

The researchers discussed a research problem concerning the logical structure of the dynamic capability framework. The team delineated the research boundaries, focusing on the primary interventions of the dynamic capabilities framework, which emphasizes managerial decisions, resource and capability configurations, and strategy execution. The researchers assembled to establish a research design for the development of a conceptual model and set a research agenda. Fig 1 illustrates the current status of the dynamic capabilities framework introduced by Teece [5].

### Searching SLR articles

An automated search of SLR articles was conducted using the Scopus database, focusing on specific keywords related to decision-making, complex domains, and RBV [53]. The search results were subsequently sorted with inclusion and exclusion criteria applied based on the subject areas of business, management, and accounting [53,54]. The final inclusion and exclusion process utilized literature to identify relevant SLR articles [54]. The development of a conceptual model necessitates an SLR article on decision making, complex contexts, and RBV to gather substantial insights in the form of key factors. These key factors contribute to enhancing the logical structure of the dynamic capabilities framework by addressing contextual issues, aligning treatment, and improving strategy alignment in accordance with the cynefin framework [4,55].

### Conducting an FGD

In the development of a decision-making model, qualitative data derived from a case study of SOB are essential to validate the theoretical framework concerning decision-making challenges within the banking sector. Participants for the FGD were selected in accordance with the triple helix model, which includes representatives from academia, banking industry, and government [56]. The FGDs were conducted through in-depth interviews using semi-structured questions [50. p. 444]. The FGD comprised four participants: two bank managers (head division), a bank practitioner (bank consultant), a central bank officer (serving as a government representative), and an academician (university member) [49. p. 113–122]. The FGD was structured as an in-depth interview to explore the current decision-making processes [50. p. 447] [45. p. 447]. These discussions addressed topics such as resources and capabilities, decision-making mechanisms, decision-making models, and strategic alignment. The reliability of the data was substantiated by the researchers by categorizing the reliability of the responses as suitable for developing the decision-making model [49. p. 348]. The validation of the qualitative data from the FGD employed the triangulation method, wherein the FGD results were reconfirmed with interviewees [51].

### Consulting an expert panel

An expert panel comprises individuals with specialized knowledge of decision making, banking, and strategic planning [23]. This panel analyzed the results of the FGD and integrated them with key factors identified from selected SLR articles to develop a conceptual framework [23,57]. Such a panel provides researchers with consultation on the methods, techniques, and materials pertinent to the advancement of decision-making processes [23].

### Developing a conceptual framework

Researchers have established emerging concepts for the development of decision-making models, drawing on the conceptual framework proposed by Ullah [55] and Ravitch and Riggan [58]. This involved utilizing materials from the FGD results and insights from selected SLR articles [55,59]. The STS model was employed to structure the interactions among people, infrastructure, technology, culture, processes, and goals [17,19,21,25]. The emerging concepts for

decision-making model development are categorized into four processes: (1) identification of key factors, (2) analysis of interrelated key factors, (3) formulation of concepts based on the analysis of interrelated key factors, and (4) proposing a conceptual framework for decision-making model development.

### Proposing a draft of decision-making model

The decision-making model was formulated based on principles that underpin its development, reflecting theoretical perspectives on decision-making within complex contexts [17,22,58]. The proposed draft of the decision-making model was structured into five key stages: 1) establishing six concepts of conceptual framework to guide the model's development, 2) arranging six foundational principles through a deductive approach, 3) examining each principle within the complex domain of the cynefin framework, and 4) creating an initial draft of the decision-making model. The draft was subsequently submitted to an expert panel for evaluation.

### Reviewing a draft of decision-making model

An expert panel conducted a comprehensive review of the draft decision-making model to offer insight into its enhancement. Experts meticulously scrutinized the draft and provided recommendations for improvement. In addition, both experts and researchers compiled meeting notes that included corrective actions aimed at refining the draft.

### Revising a draft of decision-making model

The researchers convened to discuss recommendations for drafting a decision-making model. They completed their tasks in accordance with meeting notes to enhance one component of the model. Each researcher worked independently and focused on the assigned tasks to refine the model. The results were subsequently synthesized during a meeting and deliberated to confirm any improvements.

### Reaching consensus

An enhanced decision-making draft has been submitted to a panel of experts. These experts and researchers deliberated on the improved drafts. Experts evaluated the decision-making model in accordance with their recommendations. A consensus will be reached among researchers and experts if the improvements align with recommendations. If the draft failed to meet the recommendations, the experts requested that the researchers revisit the model.

### Finalizing a decision-making model

Following a consensus among researchers and experts, a decision-making model was finalized. This model was established during a meeting and its approval was documented in a previous study.

## Results and discussions

### Scoping of the decision problem

The scoping of the decision problem was conducted from July 3–31, 2023. During a meeting, researchers formulated the scoping of the decision problem, emphasizing that the logical structure of the dynamic capabilities framework should address the complex context, appropriate treatment in managerial decision-making, and alignment of resource and capability configurations with strategy. Researchers have established a developmental agenda to enhance this framework by addressing these three gaps. The primary focus of model development is managerial decisions, resource and capability management, and determining resource and capability configurations. The cynefin framework was employed to refine the logical structure of the dynamic capabilities framework, encompassing these three focal areas [4,5,17,25].

**Searching SLR articles**

In August 2024, an SLR was conducted by executing an automated search of the Scopus database. The search employed three specific keywords with the prompt: "decision-making" AND "complex context," AND "resource." The automated search initially identified 25 articles. The inclusion and exclusion criteria were applied, focusing on the subject areas within Business, Management, and Accounting, which narrowed the selection to seven articles. Further refinement based on the research type, specifically literature review, resulted in the final selection of one article. The selected study, titled "Decision Making Mechanism in Resource Based Theory: A Literature Review, Synthesis, and Future Research" by Ristyawan et al. [6], represents the culmination of this search process. The inclusion and exclusion criteria applied during the SLR search are shown in Fig 3.

The final selected SLR article provides insights that utilized to improve the dynamic capabilities framework. The insights of Ristyawan et al.'s article for decision-making model development are presented in Table 1.

This SLR study resulted in the final selection of 27 articles, of which 16 elaborated on the decision-making process in complex contexts. The final SLR selection provided 22 co-occurrence keywords that delineated the key terms of the decision-making model for further study. The co-occurrence keywords of the SLR comprise the basic framework, complex context, treatments/methods, and strategy alignment for decision-making model development. The development of a decision-making model requires the basic framework to be enhanced in a complex context. This research enhances Teece's dynamic capabilities framework [5] as the basic framework. A complex context describes the contextual setting of the decision-making process. The cynefin framework delineates the managerial actions of the decision-making process in complex situations [4]. Treatments refer to the decision-making process by which decision-makers provide the optimal decision. The decision-making model development includes the methodology of BDA and MCDM to adapt to rapid strategy changes [60]. The strategy alignment demonstrates that resource and capability allocation can support business activities in a corporate environment. The alignment of strategy and resource/capability configurations can lead to a competitive advantage and enhanced performance [61].

**Conducting an FGD**

In the final week of August 2023, the research team initiated the process of scheduling the FGD and extended invitations to FGD participants via email. Upon confirmation of their attendance, the FGD was conducted on September 15, 2023. The FGD facilitated a structured interview with four stakeholders: a head division representative, bank consultant, central bank officer, and academician. The FGD session lasted for approximately two hours. The outcomes of the FGD are presented in Table 2.

The researchers employed triangulation techniques to validate the FGD, which corroborated the responses of the FGD participants. The FGD sessions were video-recorded, capturing interviews with informants. Subsequently, the researchers transcribed video recordings. In the final week of September, the data validation of the FGD results was conducted. The researchers sought confirmation from all informants regarding the accuracy of transcripts. All informants affirmed that the transcripts accurately reflected their responses to the FGD.

Furthermore, researchers have utilized the content analysis method to scrutinize insights emerging from the FGD results. These insights were aligned with the decision-making processes concerning the allocation of resources and capabilities. The content analysis, grounded in the FGD findings, highlighted several critical elements, such as the decision-making process for resource and capability allocation, integration of knowledge from all stakeholders within the bank to ensure comprehensive decision-making, capacity to navigate the complexities of environmental changes in accordance with the complex domain of the cynefin framework, and beneficial role of information technology in enhancing data processing to support optimal decision-making. All identified insights were used for conformity analyses to conceptualize interrelated key factors.

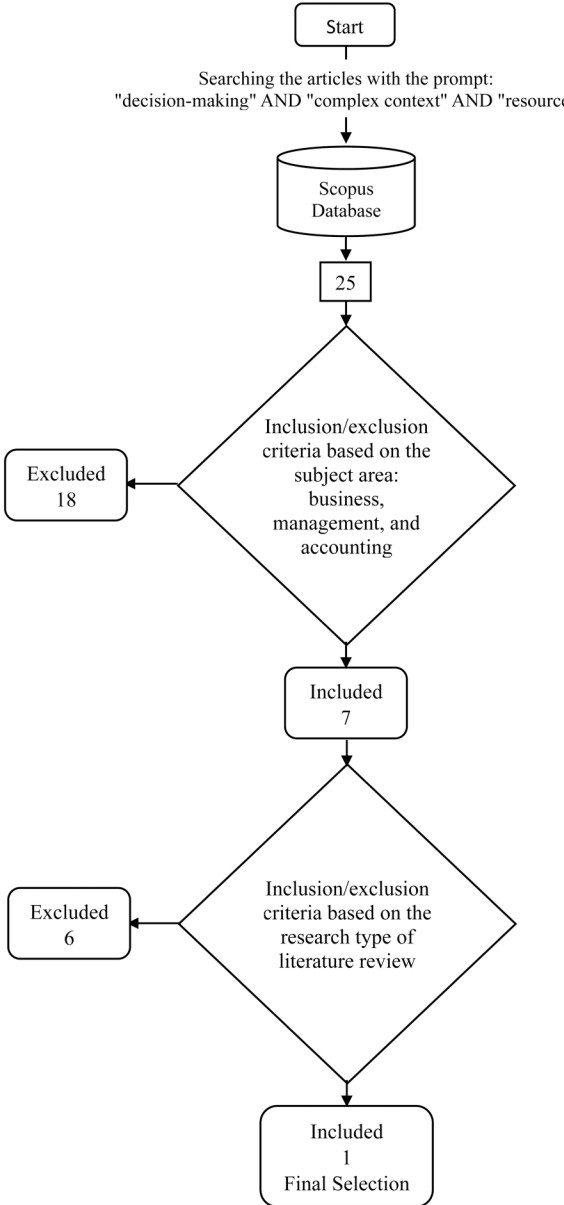

**Fig 3. Inclusion/exclusion searching SLR articles.**

## Consulting an expert panel

In early October 2023, researchers and experts convened an expert panel to discuss the results of searching for SLR articles and FGD. The researchers identified 22 key factors related to decision-making derived from the co-occurrence of keywords in the study by Ristyawan et al. [6]. These 22 key factors address RQ2 and serve as the foundation for developing a decision-making model. Additionally, the researchers presented key terms from the FGD results, which were the outputs of the content analysis. Experts have recommended employing the STS model to contextualize the results. Subsequently, an expert panel established a research agenda for the development of a conceptual framework.

**Table 1. SLR article for research agenda.**

| Item Identification | Ristyawan et al.'s article [6] | Research Agenda |
|---|---|---|
| Objectives | To find out the decision-making mechanism in RBT by reviewing and evaluating the literatures | to develop a decision-making model for determining resources and capabilities through a dynamic capabilities framework. |
| Scope of Research | Banking | Banking |
| Methodology | SLR | The development of a conceptual model by ISPOR-SMDM |
| Substantive findings on the enhancement of the dynamic capabilities framework. | 1. 27 articles expounded on the decision-making process for determining resources and capabilities that were mostly conducted in complex contexts.<br>2. This study identified 22 co-occurrence keywords that delineated the key terms of the decision-making model.<br> - Dynamic capabilities, decision making, enterprise resource management, and resource-based view present the basic framework for decision-making model development.<br> - Agility, COVID-19, emerging economies, emerging markets, and uncertainty describe the complex conditions in the decision-making context.<br> - Big data, micro-foundation, DEMATEL, big data analytics, data analytics, analytical hierarchy process, multi-criteria decision-making, and information management exhibit treatments or methods in the decision-making process.<br> - Competitive advantage, industrial management, entrepreneur, strategy, and supply chains depict terms of strategy alignment in the decision-making process.<br>3. The characteristics of decision-making mechanisms are real-time, dynamic, adaptive, and iterative, which provide insights into the design of a continuous decision-making process. | 1. Six concepts emerge from the interconnection of keywords in Ristyawan et al.'s article, which underlies model development.<br>2. This study proposes a decision-making model for determining bank resources and capabilities derived from synthesizing SLR findings and FGD results that correspond to the complex domain of the cynefin framework. |
| Future directions | The research presents the development of a decision-making model that integrates:<br>- Theoretical foundation: Dynamic capabilities<br>- Contextual setting: Complex domain within the cynefin framework<br>- Key attributes: real-time, dynamic, adaptive, and iterative.<br>- Analytical approach: Combination of BDA and MCDM | -This study presents a DSS prototype for evaluating resource and capability configurations utilizing BDA and MCDM approaches.<br>-An empirical analysis of the DSS prototype is conducted using simulation techniques. |

## Developing a conceptual framework

In mid-October 2023, researchers formulated insights into the interconnected key factors. They undertook four distinct processes to establish a conceptual framework for developing a decision-making model aimed at determining resources and capabilities, resulting in the following outcomes:

**1. *Key factor identification.*** Researchers have examined key factors using the STS model [19,21] and the concept of key-term conformity [55]. The notion of conformity serves as a basis for validating these key factors. The researchers categorized the 22 key factors into six dimensions of the STS model. Subsequently, they identified the conformity between these key factors and associated keywords. Table 3 presents the identification of the key factors analyzed in accordance with the STS model and key term conformity.

The STS model categorizes the key factors into six dimensions. According to [21], these dimensions are identified as people, infrastructure, technology, culture, processes/procedures, and goals. Within the people dimension, two key factors were identified: micro-foundation and entrepreneurship. The infrastructure dimension includes two key factors: enterprise resource management and information management. The technology dimension comprises three key factors: big data, big data analytics, and DEMATEL. Furthermore, the cultural dimension encompasses a resource-based view, COVID-19, industrial management, agility, emerging economies, emerging markets, and uncertainty. The processes/procedures dimension is characterized by seven key factors: dynamic capabilities, decision-making, analytical hierarchy process, data analytics, multi-criteria decision-making, strategy, and supply chains. Competitive advantage is identified as the goal

**Table 2. The outlines resume of FGD.**

| Member of FGD | Outlines of Interview |
|---|---|
| Head Divisions | They reported that the decision-making process to determine resources and capabilities in their bank aligned with Teece's dynamic capabilities framework. They described the decision-making process in their bank, which was conducted through annual meetings involving the BOD, division heads, and branch heads. The dynamic capabilities framework functioned effectively under normal conditions where factors, actors, and patterns were identifiable. A top-down decision-making approach is feasible under these circumstances. However, the COVID-19 pandemic has rendered this framework approach ineffective for configuring resources and capabilities. The BOD encountered limited access to information during the decision-making process. They recognized the necessity for a heuristic decision-making model that incorporated knowledge from all parties in the bank to provide comprehensive decisions. |
| Bank Consultant | He posited that the dynamic capabilities framework was a normative linear model. The real-world business environment represents dynamic circumstances. Banks require a dynamic and iterative model to adapt to rapid environmental change. Consequently, he asserted that information technology and digital systems play a significant role in assisting banks with operational activities, particularly in the decision-making process to determine resources and capabilities. He contended that the dynamic capabilities framework needs enhancement in managerial decision-making. This model should incorporate big data technology to improve decision quality. Furthermore, the model requires the development of methodologies that encompass information from all departments and divisions. |
| Central Bank Officer | He indicated that banks encountered significant economic challenges during the COVID-19 pandemic. Decision-making has emerged as a pivotal factor in preparing resources and capabilities for new strategic initiatives. He explained that banks could utilize the dynamic capabilities framework to address the complex situations caused by the pandemic. Nonetheless, he acknowledged that the decision-making process was more intricate than the framework implied. This framework requires a detailed selection mechanism. The central bank allows banks to develop their decision-making models, provided they comply with banking regulations such as the integration of data technology. He further emphasized that developing decision-making models that incorporate information technology is advantageous as it enhances data processing to support optimal decision-making. The central bank's role is to ensure that financial institutions such as banks conduct their business operations efficiently. |
| Academician | He asserted that banks play a crucial role in the national economy. Consequently, bank management must exhibit agility when adapting to a dynamic environment. The decision-making process should be flexible and accommodate changes in the surrounding circumstances with the involvement of all relevant parties. The dynamic capabilities framework should address the complexity of environmental changes in accordance with the complex domain of the cynefin framework. The BOD should investigate problem situations, discern cause-effect patterns, and respond with appropriate managerial actions. These managerial actions should clearly delineate how to structure and integrate resources and capabilities, and form a configuration to implement the strategy effectively. |

dimension within the decision-making mechanism. The identification of these key factors facilitated the organization of interrelated key factor analysis. The dimensions of the STS model, along with the key term conformity, provided valuable insights for interpreting the 22 key factors in the interrelated key factor analysis.

2. ***Interrelated key factor analysis.*** The researchers identified the interrelated key factors considered within both the STS model and the cynefin framework [4,19]. These key factors were synthesized using the STS model framework to construct the concepts of decision-making model development [21]. The cynefin framework was employed as a theoretical basis to delineate the decision-making context based on these key factors [17]. The STS model adapted six dimensions to provide interrelated key factors addressing RQ3. Fig 4 presents an analysis of interrelated key factors within the six dimensions of the STS model, offering insights into the principles of decision-making model development. The infrastructure dimension underscores the resource decision-making model, decision support system, and accelerated decision-making process facilitated by information, communication, and technology (ICT). The processes/procedures dimension includes resource orchestration and dynamic capabilities, participatory decision-making processes, hierarchical processes, resource data, VRIN criteria, strategy execution, and supply chain processes.

Subsequently, the people dimension refers to managerial decisions involving probing action, accumulation of decision-makers' knowledge and experience, and decision-makers' judgment skills with ICT. The cultural dimension emphasizes resources as a key factor, an unstable complex environment, the integration of resources, people, and technology, a lack of decision-making agility, dynamic macro-economic conditions, sudden market changes, and immediate

**Table 3. Key Factor Identification.**

| Key Factor | Dimension of STS model | Key term conformity |
|---|---|---|
| dynamic capabilities | processes/procedures | accommodate changes |
| decision making | processes/procedures | annual meeting |
| enterprise resource management | buildings/infrastructures | resource and capability allocation |
| resource-based view | culture | resources and capabilities |
| big data | technology | data technology |
| competitive advantage | goals | strategy goals |
| COVID-19 | culture | complex situations |
| industrial management | culture | bank management |
| micro-foundation | people | detailed selection mechanism |
| agility | culture | flexible |
| analytical hierarchy process | processes/procedures | decision-making approach |
| big data analytics | technology | data technology |
| data analytics | processes/procedures | data technology |
| dematel | technology | decision-making approach |
| emerging economies | culture | dynamic environment |
| emerging markets | culture | dynamic environment |
| entrepreneur | people | managerial actions |
| information management | buildings/infrastructures | information technology |
| multi-criteria decision-making | processes/procedures | decision-making approach |
| strategy | processes/procedures | bank's strategy |
| supply chains | processes/procedures | operational activities |
| uncertainty | culture | dynamic environment |

decisions. Moreover, the technology dimension encompasses big data infrastructure; BDA in the decision-making process; and real-time, adaptive, dynamic, and iterative decisions. Finally, the goal dimension highlights a competitive advantage, particularly in decision-making models that enable the BOD to promptly provide resource and capability configurations.

3. *Formulation of six concepts based on the analysis of interrelated key factors.* Researchers have established six concepts for the development of a decision-making model by analyzing interrelated key factors through the relationships among various dimensions. These concepts are informed by the work of prominent scholars who have conducted research on RBV, heuristic models, decision-making contexts, MCDM data-driven technology, and expert preference judgment [53]. References were meticulously selected by considering the most frequently cited sources [53]. Table 4 illustrates the formulation of the concepts based on the relationships among these dimensions. The first concepts was established based on the interactions between the goal dimension and other dimensions, which facilitated the acceleration of the decision-making processes for determining resources and capabilities. The theoretical foundation of this concept posits that a firm must dynamically and swiftly prepare its resources and capabilities to execute a strategy in a complex environment. This concept is consistent with the findings of Ristyawan et al. and the results of FGDs, which suggest that decision-making mechanisms for determining resources and capabilities should be real-time, dynamic, adaptive, and iterative in alignment with dynamic environmental conditions.

Furthermore, the second concept arises from the interrelations between the processes/procedures dimension and other dimensions, which formalizes heuristic judgment mechanisms. Drawing from the literature, this concept underscores that the heuristic model of the decision-making process necessitates the involvement of all parties within a company to facilitate comprehensive decision making. The second concept incorporates findings from the FGD, emphasizing that decision-making should engage all stakeholders to effectively address uncertainty.

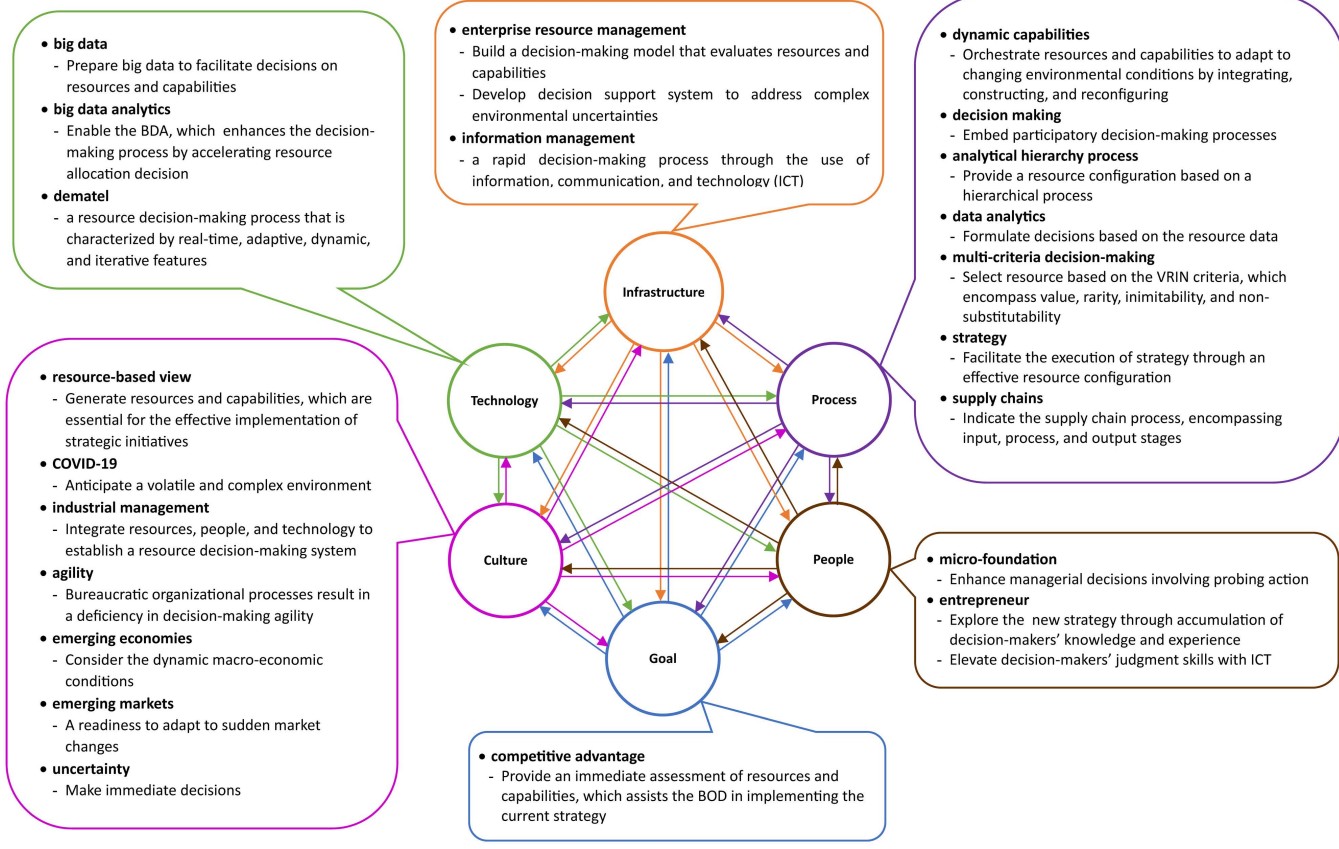

**Fig 4. Interrelated Key Factor Analysis within Six Dimensions of STS Model.**

The third concept emerges from the interplay between culture and other dimensions that creates a complex context for decision-making. According to foundational theories of thought, this concept illustrates that decision making frequently occurs within an uncertain and highly complex environment. This concept is supported by the findings of Ristyawan et al. and the outcomes of FGD, which indicate that corporate decision-making issues are situated within intricate environmental contexts.

Moreover, the fourth concept results from the relationships between the technology dimension and other dimensions, conceptualizing big data analytics for rapid decision-making processes. Based on theoretical thought, this concept emphasizes that nonlinear data-driven technology accommodates a large data-processing velocity to support a heuristic model of decision-making processes involving all managers. The fourth concept aligns with both Ristyawan et al.'s findings and the FGD results, indicating that data processing technology enforces decision-making processes.

The fifth concept was formulated by researchers in compliance with the relationship between the technology and other dimensions. Based on the literature, the fifth concept highlights deep learning with fuzzy preference relationships. This concept is related to the finding of Ristyawan et al. that real-time, dynamic, adaptive, and iterative decisions require technology-based decision-making.

Finally, the sixth concept emerged from the results of the relationship analysis between people and other dimensions. The theoretical rationale emphasizes that decision makers utilize their knowledge to consider alternatives for the best decision. Both Ristyawan et al.'s findings and the FGD results underpin the sixth concept, indicating that managers employ their knowledge to make judgments in decision-making.

Table 4. Formulation of concepts for development of decision-making model.

| Interrelationship of one dimension with others | Theoretical rationale derived from the analysis of dimensional relationships | Concepts for developing decision making model |
|---|---|---|
| Goals ◊ Processes/ Procedures<br>Goals ◊ Culture<br>Goals ◊ Technology<br>Goals ◊ Infrastructures<br>Goals ◊ People | A firm strategically allocates resources and develops capabilities in a dynamic and expeditious manner to effectively implement its strategy in a complex environment [1,3,16,31,62]. | Acceleration of the decision-making processes for determining resources and capabilities |
| Processes/ Procedures ◊ Culture<br>Processes/ Procedures ◊ Technology<br>Processes/ Procedures ◊ Infrastructures<br>Processes/ Procedures ◊ People | The heuristic model of the decision-making process engages all stakeholders within the organization to facilitate the formulation of comprehensive decisions [8,9,37,46,47]. | Heuristic judgment mechanisms |
| Culture ◊ Infrastructures<br>Culture ◊ People<br>Culture ◊ Goals<br>Culture ◊ Processes/ Procedures | Decision-making frequently occurs in environments that are characterized by uncertainty and high complexity [4,10,16,37]. | Complex context of decision making |
| Technology ◊ Infrastructures<br>Technology ◊ Goals<br>Technology ◊ Processes/ Procedures<br>Technology ◊ Culture | Nonlinear data-driven technology facilitates rapid data processing to enhance the decision-making processes in complex environments [60,63–65]. | Big data analytics for rapid decision-making processes |
| Infrastructures ◊ People<br>Infrastructures ◊ Processes/ Procedures<br>Infrastructures ◊ Culture<br>Infrastructures ◊ Technology | A data technology application facilitates MCDM, preference assessments, heuristic modeling, and predictive analysis to manage complexity effectively [37,45–47,66,67]. | Deep learning with fuzzy preference relations |
| People ◊ Goals<br>People ◊ Processes/ Procedures<br>People ◊ Culture<br>People ◊ Technology<br>People ◊ Infrastructures | Decision makers utilize their expertise to evaluate various alternatives to arrive at the optimal decision [4,8,9,31,37,45,46]. | Decision-makers' preference judgements |

**4. *Proposing a conceptual framework.*** Researchers have identified six concepts derived from the analysis of key interrelated factors. These concepts are as follows: 1) the acceleration of decision-making processes for determining resources and capabilities, 2) heuristic judgment mechanisms, 3) the complex context of decision-making, 4) big data analytics for rapid decision-making processes, 5) deep learning with fuzzy preference relations, and 6) decision-makers' preference judgments. These six concepts underpin the development of a decision-making model based on the dynamic capabilities framework.

The conceptual framework was in the form of a hexagonal network hub with a center element network. The centre element was the "Decision-Making Model for Determining Resources and Capabilities." The six concepts of decision-making model development play a role in the network hub in constructing the decision model development. Fig. 5 shows the conceptual framework for the decision-making model development. The hexagonal network hub of the conceptual framework answers RQ4 regarding the conceptual framework form for decision-making model development.

The acceleration of decision-making processes concerning the determination of resources and capabilities suggests that the development of decision-making models is intended to facilitate rapid decision-making. It is imperative for researchers to adapt the managerial decision component of the dynamic capabilities framework to ensure that the resulting decisions align with current conditions. This principle necessitates the reorganization of managerial decisions within

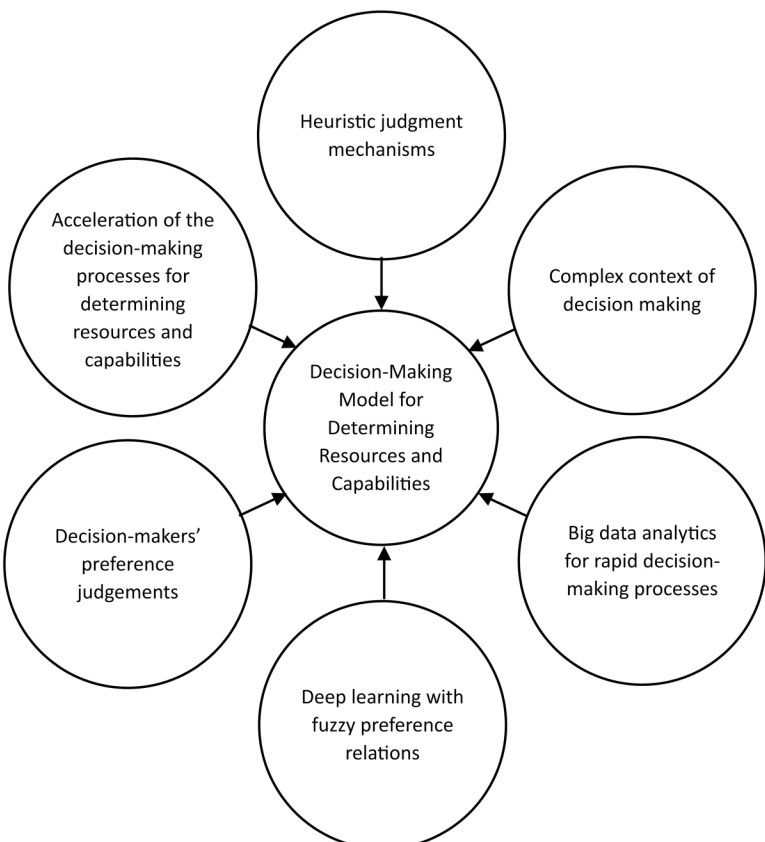

**Fig 5. Proposed conceptual framework of the decision-making model development.**

the dynamic capabilities framework to correspond with the complex domain of the cynefin framework and new strategic direction.

Heuristic judgment mechanisms offer recommendations for decision-making model development, which involves the collection of information from all parties within the bank during the decision-making process. This concept underscores the importance of investigative actions in the decision-making process of complex situations. According to the cynefin framework, decision-makers should examine cause-effect patterns when confronted with complex situations.

A complex decision-making context necessitates the creation of models that improve managerial decisions. Within the cynefin framework, the complex domain is defined by three managerial actions: "probe," "sense," and "respond." The concept of complex decision-making contexts suggests that decision-making models should incorporate these actions to ensure precise decision making. This concept clarifies the processes involved in decision making.

Furthermore, big data analytics for rapid decision-making processes complement the data processing section in decision-making model development. The big data analytics facilitate to collect information as much as possible for decision-making process to provide the accurate decisions. The big data analytics for rapid decision-making processes concept suggests the use of big data to accelerate the decision-making process.

Deep learning, when integrated with fuzzy preference relations, plays a significant role in facilitating predictive analysis in the development of decision-making models. This predictive analysis is instrumental in addressing uncertainties inherent in complex situations. The concept of deep learning combined with fuzzy preference relations offers guidance for the

development of decision-making models by interpreting the patterns of preference judgments from all stakeholders within a banking context, thereby aiding the determination of resources and capabilities.

Finally, decision-makers' preference judgments constitute the essential knowledge required for the formulation of decisions. These judgments function as inputs for the construction of the decision-making models. The concept of decision-makers' preference judgments recognizes the participatory nature of the decision-making process, wherein the allocation of resources and capabilities is harmonized with the requirements of all divisions and departments during the execution of strategy.

### Proposing the draft of decision-making model

In the first week of November 2024, the researchers initiated the development of a draft decision-making model. This process involves four distinct stages. The outcomes of the decision-making model development are as follows:

**1) *Establishing six foundational concepts to guide the model's development.*** The six concepts of conceptual framework were systematically organized in an analytic table that included six columns dedicated to the concepts. Additionally, the table was extended to incorporate improvements to the dynamic capabilities framework and the steps involved in the model elements. The enhancements to the dynamic capabilities framework involve restoration to facilitate the development of a decision-making model. The procedural steps within the model elements outline the decision-making processes that ultimately determine the configuration of resources and capabilities.

**2) *Arranging six concepts through a deductive approach.*** Researchers have employed a deductive approach to organize the six foundational principles, arranging them from general to specific contexts. Initially, they identified a general principle that serves as the foundational layer for model development. This principle concerning the acceleration of decision-making processes for resource and capability determination was established as the basic analytical layer.

Subsequently, the researchers designated the complex context of decision making as the second layer. This layer outlines the first layer in relation to managerial decisions within the complex decision-making context, corresponding to the cynefin framework, which includes the stages of "probe," "sense," and "respond." The third layer involved the heuristic judgment mechanism, which elaborated on the practices of the three managerial actions within the banking context to address complexity.

The fourth layer focuses on decision makers' preference judgments, highlighting the role of actors in the decision-making process. This layer indicates that heuristic judgment mechanisms require decision-makers' preferences as input values to facilitate decision making. The fifth layer concept, big data analytics for rapid decision-making processes, serves as a means to process preference values from the evaluators. Big data provides extensive storage of preference data for the decision-making process. The sixth layer represents deep learning with fuzzy preference relations, elaborating on data-processing methods to predict optimal decisions based on patterns of preference values in complex situations.

Finally, researchers introduced two additional layers for decision-making development: enhancements to the dynamic capabilities framework, and procedural steps within the model elements. Improvements to the dynamic capabilities framework elucidate the advancements necessary to develop a decision-making model for resource and capability determination. The procedural steps within the model elements translate these improvements into actionable steps to make resource and capability decisions.

**3) *Examining each concept within the complex domain of the cynefin framework.*** Researchers have established a cynefin framework as a theoretical foundation for delineating six key concepts. Each concept was analyzed to derive insights into the boundaries of complex domains. These insights were subsequently integrated to identify areas for enhancing dynamic capabilities, which were then incorporated into the decision-making model. The results of the analysis of these six principles are presented in Table 5.

**Table 5. The development of the decision-making model.**

The six concepts for decision-making model development

| The acceleration of decision-making processes for determining resources and capabilities | Complex context of decision-making | Heuristic judgment mechanisms | Decision makers' preference judgements | Big data analytics for rapid decision-making processes | Deep learning model with fuzzy preference relations | The improvements of dynamic capabilities framework | The procedural steps within the model elements |
|---|---|---|---|---|---|---|---|
| Develop a decision-making model which helps the BOD provide real-time, adaptive, dynamic, and iterative decisions | *Probe* Create a decision-making scenario by investigating the patterns between resources/ capabilities and the their success rate of strategy execution | Require heuristic judgment mechanisms to synthesize all managers' judgments | The BOD identifies the data patterns of managers' preference judgments. | Trials in some experiments to investigate the patterns | Facilitate the patterns between resources/capabilities and their success rate of strategy execution | Revise *sense* of the managerial decisions with *"probe"* | 1.Probe some experiments to determine the patterns of decision-making scenarios 2.Create a decision scenario |
| | *Sense* Recognize the relationship patterns between resources/ capabilities and the their success rate of strategy execution | Establish the judgment types for resources and success rate of resources | The BOD understands the data patterns of managers' preference judgments. | Provide a large database that shows the data patterns between resources/ capabilities and their success rate | Improve the learning skill to understand patterns between resources/ capabilities and their success rate | Replace *seize* of the managerial actions with *"sense"* | 3.Diagnose the decision-making scenario to provide real time, dynamic, adaptive, and iterative decisions |
| | *Respond* Design the decision-making model with MCDM and BDA method, heuristic model, fuzzy preference relations, and a predictive analysis. | Determine methods for synthesizing resources and capabilities judgments | Managers are involved in decision-making as decision-makers. | Serve a resource and capability database | Employ a predictive analysis to structure resources and capabilities | Replace the *transform* of the managerial actions with the *respond* and outline the *respond* with the first action of *"structuring"* for a resources and capabilities configuration | 4.Conduct a group discussion to identify resources and capabilities 5.Screen resources and capabilities 6.Sort VRIN resources, dynamic capabilities, and ordinary resources/capabilities 7.A set of reliable resources and capabilities |
| | | Collect all managers' judgments for decision-making inputs | Managers give their preference judgments for resources and their success rate. | Record managers' judgments in big data storage. | Prepare big data bundling of managers' judgments as inputs for a predictive analysis. | Emerge the second action of *"bundling"* in respond to collect managers' judgments | 8.Arrange database storage for fuzzy preference relations between resources/capabilities and success rate 9.Collect managers' preference judgments |

*(Continued)*

**Table 5.** (Continued)

**The six concepts for decision-making model development**

| The acceleration of decision-making processes for determining resources and capabilities | Complex context of decision-making | Heuristic judgment mechanisms | Decision makers' preference judgements | Big data analytics for rapid decision-making processes | Deep learning model with fuzzy preference relations | The improvements of dynamic capabilities framework | The procedural steps within the model elements |
|---|---|---|---|---|---|---|---|
| | | Create a large database for all managers' judgments for serving data processing. | Managers' preference judgments constitute inputs for building up a large database. | Set big data of managers' judgments as input for decision-making process. | Build up a big data of managers' judgments as inputs for a predictive analysis. | Create the third action of *"build-ing"* in the *respond* to build a big data of managers' judgments | 10.Build up a large database of managers' judgments 11.Prepare inputs of fuzzy preference relations for predictive analysis 12.Run deep learning analytics—predictive analysis of success rate |
| | | Utilize the predictive analysis results of judgment synthesis to formulate a resource and capability configuration based on the success rate of strategy execution | The BOD uses the data processing results of managers' preference judgments to determine a resource and capability configuration. | Provide the results in the form of a success rate for each resource and capability through deep learning analysis. | Process the big data of managers' judgments to provide a predictive analysis for success rate of resources/capabilities | Create the fourth action of *"leverag-ing"* in the *respond* to harness the results of predictive analysis to build a resources/capabilities configuration to execute the strategy | 13.Formulate a resources and capabilities configuration 14.Deploy a resources and capabilities configuration for the strategy execution |
| | | Reconstruct a judgments database based on resource and capability reconfiguration | The BOD sorts the resources and capabilities that do not provide a success rate and sends them to the remedial process. | restructure database of managers' judgments | Send resources/capabilities with no success rate of to a reconfiguration phase | Create the fifth action of *"recon-figuring"* in the *respond* to evaluate the resources and capabilities configuration | 15.Set a remedial action |

First, the acceleration of decision making in determining resources and capabilities serves as a foundational element for developing decision-making models. This offers guidance for model development, emphasizing the need for real-time, adaptive, dynamic, and iterative decision-making processes. This concept aligns with the complex context of the cynefin framework and Teece's dynamic capabilities, suggesting that decision-making regarding resources and capabilities should be responsive to environmental conditions [3–5,17]. Consequently, this decision-making model must be efficient in terms of both decision time and quality to integrate, build, and reconfigure a company's resources and capabilities effectively.

Second, the complex context of resource decision making highlights critical aspects of Teece's dynamic capabilities framework, indicating that this model should adhere to the clauses of the complex domain within the cynefin framework. The complex domain involves actions such as probing, sensing, and responding [4,17]. Criticisms of the dynamic capabilities framework suggest that managerial decisions often remain within a complicated context characterized by "sense," "seizing," and "transform" [5,9]. By contrast, the complex domain of the cynefin framework posits that managerial decisions should commence with "probe," "sense," and "respond." Therefore, this study revises managerial decisions within the dynamic capabilities framework to align with the cynefin framework. Table 5 presents the insights related to "probe", "sense", and "respond." The "probe" involves investigating the patterns between resources/capabilities and the success rate of strategy execution. The "sense" involves recognizing the relationship patterns between resources/capabilities and the success rate of strategy execution. Finally, "respond" involves the development of a decision-making model.

The third concept concerns to heuristic judgment mechanisms that outline "probe," "sense,'" and "respond" of managerial actions of decision-making processes [17,68]. These three actions within the complex domain form the basis for seven insights into the heuristic judgment mechanisms. The first insight, represented by the "probe," suggests that a decision-making model requires heuristic judgment mechanisms to synthesize all managerial judgments. The second insight, reflecting this "sense," indicates that a decision-making model establishes judgment types for resources and the success rate of resources/capabilities. "Respond" represent the third to seventh insights. The third insight emphasizes method selection for synthesizing judgments regarding resources and capabilities. The fourth insight involves the collection of resource and capability judgments from all managers. The fifth insight underscores the need to create a comprehensive database for processing managerial judgments. The sixth insight highlights the use of predictive analysis results from judgment synthesis to formulate a resource and capability configuration based on the success rate of the strategy execution. The final insight involves reconstructing a judgment database based on resource and capability reconfiguration.

Moreover, the fourth concept pertains to decision-makers' preference judgments, which reflect human behavioral involvement in the decision-making process. This principle includes the participation of the BOD and managers. The BOD plays a role in examining the patterns of decision-making scenarios, whereas managers offer preference judgments regarding resources and capabilities during the decision-making process. These preference judgments align with Daniel Kahneman's notion of value judgment, wherein experts provide their assessments of all decision alternatives [46]. Table 5 presents seven insights derived from the heuristic judgment mechanisms. The first, second, sixth, and seventh insights suggest that BOD identifies and comprehends the data patterns of managers' preference judgments, leading to the configuration of resources and capabilities. The third to fifth insights highlight managers' roles in sharing preference judgments regarding resources and capabilities.

The fifth concept involves big data analytics for a rapid decision-making process, which necessitates database technology to deliver effective information in terms of time and quality. The seven insights related to big data analytics for rapid decision-making indicate that big data analytics functions as a substantial database for the complex context of decision-making. Big-data analytics facilitates experimental trials to investigate these patterns. This concept posits that big data analytics constructs a large database for predictive analysis and elaborates on the role of big data as an input that influences the success rate for each resource and capability.

The final concept pertains to deep-learning analytics with fuzzy preference relations. Deep learning analytics conducts a predictive analysis to ascertain the success rate of each resource and capability. Fuzzy preference relations reveal

relationship patterns between resource and capability preferences and success rate preferences. The deep learning model aids in pattern investigation, enhances learning skills for understanding patterns, and serves as a predictive analysis tool for determining the success rate of resources and capabilities.

The improvement layer suggests restructuring actions for managerial decisions within the dynamic capability framework. The first improvement involves revising the sense of managerial decisions with the "probe." The second improvement entails replacing the "seize" of managerial actions with "sense." The third improvement involves substituting the "transform" of managerial actions with 'respond' and outlining the 'respond' with the initial action of "structuring" for a resources and capabilities configuration. The fourth improvement introduces the second action of "bundling" in the "respond" to gather managers' judgments. The fifth improvement involves creating the third action of "building" in the "respond" to develop a big data repository of managers' judgments. The sixth improvement involves establishing the fourth action of "leveraging" in the 'respond' to utilize the results of predictive analysis to construct a resources/capabilities configuration for strategy execution. The final improvement involves creating the seventh action of "reconfiguring" in the 'respond' to evaluate the configuration of resources and capabilities.

The final layer of decision-making model development delineates the procedural steps of the components of the model. Steps 1 and 2 involve the initial probing phase, which entails conducting experiments to discern patterns in decision-making scenarios and subsequently creating a decision scenario. Step 3 pertains to the diagnostic phase, which involves analyzing the decision-making scenario to facilitate real-time, dynamic, adaptive, and iterative decision-making processes. Steps 4–7 pertain to the structuring phase, which includes conducting group discussions to identify resources and capabilities; screening these resources and capabilities; categorizing VRIN resources, dynamic capabilities, and ordinary resources/capabilities; and preparing a comprehensive set of resources and capabilities. Steps 8 and 9 involve the bundling phase, which includes organizing database storage for fuzzy preference relations between resources/capabilities and success rates and gathering managers' preference judgments. Steps 10–12 pertain to the building phase, which involves constructing an extensive database of managers' judgments, preparing inputs of fuzzy preference relations for predictive analysis, and executing deep learning analytics for the predictive analysis of success rates. Steps 13 and 14 involve the leveraging phase, which includes formulating a configuration of resources and capabilities, and deploying this configuration for strategic execution. The final step involves the reconfiguring phase, which involves setting a remedial action.

**4) *Creating an initial draft of the decision-making model.*** Researchers have synthesized insights into dynamic capability enhancements and integrated them into the components of the proposed decision-making model. This model comprises four primary elements: managerial decisions, strategy, competitive advantage, and performance. Managerial decisions are categorized into three actions: "probe," "sense," and "respond." The "respond" action is further divided into five sub-actions: structuring, bundling, building, leveraging, and reconfiguring. The steps within the model's elements elucidate the processes of "probe," "sense," "structuring," "bundling," "building," "leveraging," and "reconfiguring." These 15 steps detail the actions and sub-actions of the managerial decisions. A draft of the decision-making model is illustrated in Fig 6, depicting a dynamic decision-making process that begins with managerial actions ("probe," "sense," "structuring," "bundling," "building," "leveraging," and "reconfiguring"), followed by strategy, competitive advantage, and performance. Managerial actions describe how the BOD probes and senses patterns in resource/capability and success rates concerning an issue.

Subsequently, the BOD structures the configuration of resources and capabilities and gathers managers' preferences regarding these resources and capabilities. Additionally, BOD constructs a comprehensive database of fuzzy preference relations (the relationship between resource/capability preferences and success rate preferences) to predict the success rate of resources/capabilities. BOD leverages the results of the predictive analysis to determine the success rate for each resource or capability. Resources and capabilities that do not yield a success rate are reconfigured. BOD formulates a configuration of resources and capabilities to implement the strategy. The success rate of resources and capabilities enables a bank to achieve competitive advantage, thereby enhancing its performance. A bank's performance

**Fig 6. The draft of the decision-making model for determining banks' resources and capabilities.**

can be assessed through managerial decisions regarding its resources and capabilities. Dynamic illustrations of the decision-making model draft address RQ5, which pertains to the description of the proposed decision-making model for determining resources and capabilities.

## Reviewing a draft of decision-making model

In mid-November 2023, researchers submitted a draft of the decision-making model to an expert panel for evaluation. The experts subsequently reviewed the draft, focusing on two key aspects: technical and conceptual. The technical review aimed to elucidate the flow of managerial decisions within the decision-making model as well as the associated validation process. Table 6 presents the results of this technical review.

Based on the results of the technical review, a draft decision-making model can be implemented to ascertain the resources and capabilities. Experts have indicated that this model is suitable for assisting the BOD in addressing issues related to resource and capability configuration. In addition, a review of the conceptual aspect identified an incomplete step in the reconfiguration process. Experts have recommended enhancing the details of "reconfiguration" by setting remedial actions to release, replace, develop, or acquire resources and capabilities.

## Revising a draft of decision-making model

Researchers began revising the draft of the decision-making model in the last week of November 2023, and completed the first week of December 2023. Researchers have repaired a the font size of managerial decisions, strategy, competitive advantage, and performance. Furthermore, researchers complemented the *reconfiguring* with set remedial action to release, acquire, develop, or replace resources and capabilities.

## Reaching consensus

In mid-December 2023, researchers and experts convened to reach consensus on the proposed draft improvements. The researchers indicated that the revisions were conducted in accordance with experts' recommendations. The

**Table 6. The flow the decision-making model justification review by experts.**

| Managerial decisions | Input | Process | Output | Validation |
|---|---|---|---|---|
| *Probe* | Qualitative data: resource configuration in complex context | A group discussion: BOD and experts | Qualitative data: A cause-effect decision pattern between resources configuration and successful strategy execution | Confirmatory triangulation [51] |
| *Sense* | Qualitative data: A cause-effect decision pattern between resources configuration and successful strategy execution | A group discussion: BOD and experts | Qualitative data: A pattern between resources/ capabilities and their success rate | Confirmatory triangulation [51] |
| *Structuring* | A pattern between resources/ capabilities and their success rates | A group discussion: BOD and experts | Fuzzy preference relations between resources/capabilities and their success rates | Confirmatory triangulation [51] |
| *Bundling* | Fuzzy preference relations between resources/capabilities and their success rates | Survey | Resources/capabilities and success rate preference judgments | Consistency ratio [44] |
| *Building* | Resources/capabilities and success rate preferences | Deep learning analytics: predictive analysis with 80% data learning and 20% data testing | Success rate for reach resource and capability | Accuracy level [69] |
| *Leveraging* | Success rate for reach resource and capability | Sorting process | •successful resources and capabilities •unsuccessful of resources and capabilities | Success: success rate > 50% No success: success rate ≤ 50% [70] |
| *Reconfiguring* | unsuccessful of resources and capabilities | • Release • Replace • Develop • Acquire | A pattern between resources/ capabilities and their success rates | The BOD judgment [9] |

experts confirmed that the improvements were appropriate and aligned with their suggestions and concluded that the decision-making model was complete.

## Finalizing a decision-making model

In a previous study, the researchers developed a decision-making model. They reported that the final decision-making model for evaluating resources and capabilities was completed on December 18, 2023. This model was assessed through a case study of an Indonesian SOB to validate its applicability. Fig 7 shows the final version of the proposed decision-making model.

## A sample case study of SOB in Indonesia

The proposed decision-making model for assessing resources and capabilities is applicable to Indonesian SOB. This model effectively organizes the resources and capabilities to establish coherent configurations. This case study further elucidates how challenges related to contextual mismatch, inappropriate treatment, and strategic alignment can be addressed. Fig 8 illustrates the implementation of the proposed decision-making model within the Indonesian SOB.

Initially, the Indonesian SOB formulated a novel strategy to address the complex environmental changes. The BOD undertook measures to prepare the resources and capabilities necessary to implement this strategy. The first phase involved probing the decision-making scenario, which encompasses the cause-effect relationships between resource and capability configurations and the strategic alignment of the SOB.

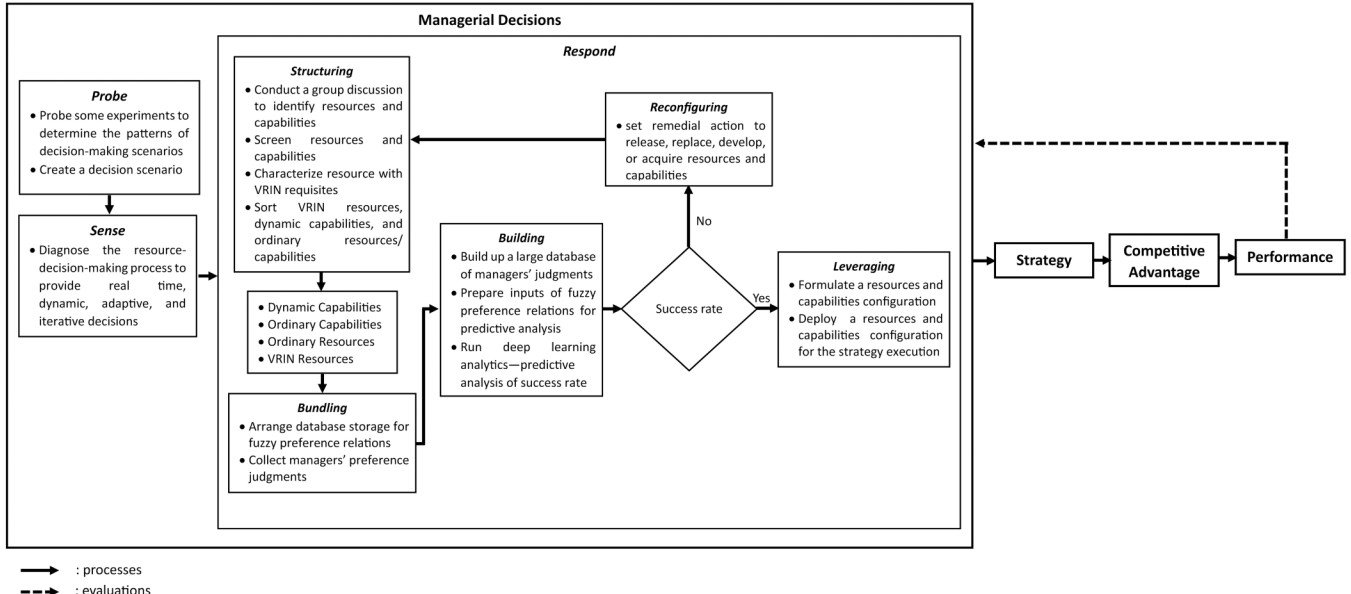

**Fig 7. The final of the proposed decision-making model for determining resources and capabilities for banks.**

Subsequently, in the second phase, the BOD established a decision-making pattern linking resources and capabilities to their success rates, recognizing these as essential assets for executing the new strategy. The success rate of the strategy execution serves as an indicator of strategic alignment.

In the third phase, a resource and capability structure was delineated during a BOD meeting, identifying ten resources and capabilities deemed suitable for executing the new strategy. These resources include human capital (HC), financial capital (FC), good corporate governance (GCG), risk management (RM), brand power (BP), international digital networks (IDN), global customers (GC), IT infrastructure (IT), digital business models (DBM), and organizational agility (OA). BOD further examined their characteristics, categorizing them as dynamic capabilities, VRIN resources, and ordinary resources/capabilities.

In the fourth phase, the BOD developed a structure of fuzzy preference relations, wherein decision-making involves preferences for resources/capabilities and success rates. These fuzzy preference relations assist BOD in establishing a database structure for decision-making analysis.

In the fifth phase, the BOD prepared an input-process-output framework for data analysis. The input for the decision-making analysis comprises preferences for the ten resources/capabilities and ten success rate preference databases. Data processing employs deep learning analytics for the predictive analysis of success rates. The output of this analysis is the predicted success rate of the ten resources, allowing the BOD to rank resources/capabilities based on success predictions.

In the sixth phase, the BOD selects resources/capabilities with success rates exceeding 50% and configures them for strategy execution, whereas those with success rates of 50% or less are earmarked for reconfiguration. The outputs of deep learning analytics, predictive analysis, identified six resources/capabilities for configuration (HC, FC, GCG, RM, IT, and OA), two for development (BP and IDN), and two for release (GC and DBM). Meanwhile, the BOD leveraged six resources/capabilities to implement the new strategy. Successful strategy execution enables the SOB to attain a competitive advantage, which in turn enhances its performance.

Ultimately, this case study demonstrates that the decision-making model effectively addresses the three gaps of mismatched context, inappropriate treatment, and strategy alignment in rapidly changing complex environments. This model

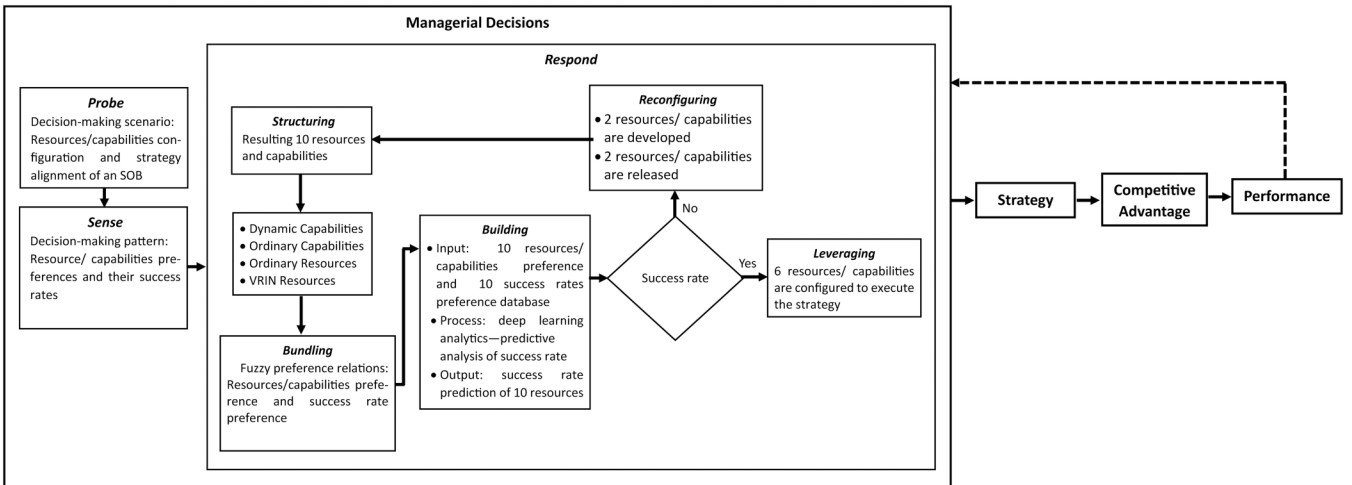

**Fig 8. Implementation of the proposed decision-making model in an SOB.**

**Table 7. Roadmap for future research.**

| Item of research | Existing research | First further research | Second further research |
|---|---|---|---|
| Research type | Qualitative research | Quantitative research | Quantitative research |
| Research topic | Conceptual model | Simulation research | Experimental research |
| Methods | Literature review and Interview | Fuzzy AHP-ANN | Fuzzy AHP-ANN |

facilitates real-time adaptive dynamic iterative resource/capability configurations in complex contexts. The success rate is indicative of the level of strategic alignment of resources/capabilities.

## Conclusions

This study enhances the dynamic capabilities framework to address three specific gaps: mismatched context, inappropriate treatment, and strategy alignment. The primary focus of improvement lies in the development of a decision-making model, particularly concerning managerial decisions. The complex domain of the cynefin framework offers guidance for addressing mismatched contexts. Detailed managerial actions have been proposed to resolve issues related to inappropriate treatment. Strategy alignment is managed through success rate preferences, which illustrate the relationship between resources, capabilities, and strategies.

In addition, this study introduced a conceptual framework derived from an analysis of keywords from a systematic literature review of SLR articles and the results of an FGD. The proposed conceptual framework is structured as a hexagonal network hub, wherein six concepts are interconnected with the central decision-making model to determine resources and capabilities. The network hub facilitated the establishment of relationships between these six concepts and the decision-making model. These six concepts include the acceleration of decision-making processes for determining resources and capabilities, heuristic judgment mechanisms, the complex context of decision-making, big data analytics for rapid decision-making processes, deep learning with fuzzy preference relations, and decision makers' preference judgments..

In addition, this study introduced a conceptual framework derived from an analysis of keywords from a systematic literature review of SLR articles and the results of an FGD. The proposed conceptual framework is structured as a hexagonal network hub, wherein six concepts are interconnected with the central decision-making model to determine resources and capabilities. The network hub facilitated the establishment of relationships between these six concepts and

the decision-making model. These six concepts include the acceleration of decision-making processes for determining resources and capabilities, heuristic judgment mechanisms, the complex context of decision-making, big data analytics for rapid decision-making processes, deep learning with fuzzy preference relations, and decision makers' preference judgments.

The proposed decision-making model demonstrates an enhancement in managerial decision-making processes. The three traditional actions of managerial decision-making within the dynamic capabilities framework—sense, seize, and transform—are supplanted by the complex domain of the cynefin framework, which includes the actions of "probe," "sense," and "respond." The "respond" action is further delineated into five specific actions: structuring, bundling, building, leveraging, and reconfiguring. These five actions characterize a decision-making process that is real-time, adaptive, dynamic, and iterative.

A case study involving an Indonesian SOB revealed that the proposed decision-making model is effective for navigating complex environmental changes. This model elucidates the progression of the decision-making process from probing to sensing, structuring, bundling, building, leveraging, and reconfiguring, thereby demonstrating its efficacy. The model successfully categorizes ten resources into six resources/capabilities configured, two resources/capabilities developed, and two resources/capabilities released.

The findings of this research contribute to theoretical advancements by enhancing Teece's dynamic capabilities framework. Managerial actions within the dynamic capabilities framework evolve in accordance with the complex domain of the cynefin framework. The proposed decision-making model aligns with Eisenhardt and Martin's assertion that dynamic capabilities must accommodate both high and moderate market velocity. Furthermore, the practical implications of this research indicate that the proposed decision-making model for resource and capability determination can assist BOD in formulating resource configurations in rapidly changing environments. Unpredictable changes pose challenges for BOD in selecting resources and capabilities for implementing new strategies.

Finally, the application of the proposed model demonstrates its effectiveness in a case study of an Indonesian SOB, showing that the model can address three resource decision-making challenges within an SOB. Therefore, the proposed decision-making model can maintain and enhance resources and capabilities during complex environmental changes.

## Limitations

This study developed a decision-making model aligned with the complex domain of the cynefin framework. The development of the decision model is centered on managerial decisions within the dynamic capabilities framework. The research domain encompasses decision-making within the RBV and the banking sector. The study is limited to a case study of a bank in Indonesia, necessitating an expansion of the scope of the investigation to generalize the results of the conceptual framework and proposed decision-making model.

Furthermore, the theoretical advancements are confined to RBV, dynamic capabilities, decision-making, and banking. This theoretical contribution constrains managerial decision enhancement in complex contexts. This study examined the subject of conceptual model development.

## Recommendations

Empirical research has demonstrated that the proposed decision-making model is effective for adapting to complex environmental changes. Therefore, it is essential to employ appropriate simulation techniques to ensure accurate results. A mixed-methods approach should be used to integrate both qualitative and quantitative data analyses.

Furthermore, the theoretical development of the proposed decision-making model needs to be advanced. This model can be enhanced through strategic alliances, resource outsourcing, and innovation in resources and capabilities. Finally, as the proposed model serves as a prototype for determining resources and capabilities, it is necessary to implement improvements at a detailed technical level.

## Future research

The findings of this study are anticipated to stimulate further research on a decision support system (DSS) aimed at determining resources and capabilities. The empirical investigation of DSS will encompass the development of decision-support tools to assist the BOD. A case study simulation of the DSS model was also conducted to evaluate its effectiveness.

Subsequently, further empirical research will be undertaken in the form of experimental studies. These experimental studies are expected to provide insights for the enhancement of an existing decision-making model. Table 7 presents the prospective research directions for the development of decision-making processes.

## Supporting information

**S1 Fig. The Comparison between complicated and complex context in dynamic capabilities framework.**
(PDF)

**S2 Fig. Research Methodology of Conceptual Model Development.**
(PDF)

**S3 Fig. Inclusion/exclusion searching SLR articles.**
(PDF)

**S4 Fig. Interrelated key factor analysis within six dimensions of STS model.**
(PDF)

**S5 Fig. Proposed conceptual framework of the decision-making model development.**
(PDF)

**S6 Fig. The draft of the decision-making model for determining banks' resources and capabilities.**
(PDF)

**S7 Fig. The final of the proposed decision-making model for determining resources and capabilities for banks.**
(PDF)

**S8 Fig. Implementation of the proposed decision-making model in an SOB.**
(PDF)

## Acknowledgments

We thank the reviewers for their insightful comments on the development of this manuscript. We also thank the School of Business and Management of Bandung Institute of Technology for supporting this research.

## Author contributions

**Conceptualization:** Utomo Sarjono Putro, Manahan Siallagan.

**Data curation:** Mochammad Ridwan Ristyawan.

**Formal analysis:** Utomo Sarjono Putro, Manahan Siallagan.

**Investigation:** Mochammad Ridwan Ristyawan, Manahan Siallagan.

**Methodology:** Mochammad Ridwan Ristyawan.

**Software:** Manahan Siallagan.

**Supervision:** Utomo Sarjono Putro, Manahan Siallagan.

**Validation:** Manahan Siallagan.

**Writing – original draft:** Mochammad Ridwan Ristyawan.

**Writing – review & editing:** Mochammad Ridwan Ristyawan.

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
