## [Decision Letter · Decision Letter 0]

10 Nov 2024

PONE-D-24-23190Determining  Resources and Capabilities in Complex Context: A Decision-Making Model for BanksPLOS ONE

Dear Dr. Ristyawan,

Thank you for submitting your manuscript to PLOS ONE. After careful consideration, we feel that it has merit but does not fully meet PLOS ONE’s publication criteria as it currently stands. Therefore, we invite you to submit a revised version of the manuscript that addresses the points raised during the review process.

**Dear authors**

**I hope you are doing well.**

The paper is nearest to being rejected; however, as I see some good contributions, the paper needs to be revised based on Reviewers comments and Mine. The following must be done: define the study framework in detail and specify the inclusion and exclusion of literature in detail. The abstract must be rewritten to include the methodology. Questions should be clearly stated in the introduction and then answered in detail in the SLR. Through a comprehensive review, you must design a table that shows the proposals for future research in terms of the descriptive and thematic aspects. You must provide a summary of the literature, stating the most important determinants in detail based on the literature. Re-extract the figures explicitly.

THNAK YOU. Please submit your revised manuscript by Dec 25 2024 11:59PM. If you will need more time than this to complete your revisions, please reply to this message or contact the journal office at plosone@plos.org . Please include the following items when submitting your revised manuscript:

We look forward to receiving your revised manuscript.

Kind regards,

Saddam A. Hazaea, Postdoctoral

Academic Editor

PLOS ONE

**Journal Requirements:**

We thank the reviewers for their insightful comments on the development of this manuscript. We also thank the School of Business and Management of Bandung Institute of Technology for supporting this research. We appreciate the Indonesian Ministry of Education, Culture, Research and Technology for supporting this research.

**Additional Editor Comments :**

Dear authors

I hope you are doing well

The paper is nearest to being rejected; however, as I see some good contributions, the paper needs to be revised based on comments. The following must be done: define the study framework in detail and specify the inclusion and exclusion of literature in detail. The abstract must be rewritten to include the methodology. Questions should be clearly stated in the introduction and then answered in detail in the SLR. Through a comprehensive review, you must design a table that shows the proposals for future research in terms of the descriptive and thematic aspects. You must provide a summary of the literature, stating the most important determinants in detail based on the literature. Re-extract the figures explicitly.

THNAK YOU

Reviewers' comments:

Reviewer's Responses to Questions

**Comments to the Author**

1. Is the manuscript technically sound, and do the data support the conclusions?

Reviewer #1: No

Reviewer #2: Partly

2. Has the statistical analysis been performed appropriately and rigorously? 

Reviewer #1: N/A

Reviewer #2: Yes

3. Have the authors made all data underlying the findings in their manuscript fully available?

Reviewer #1: Yes

Reviewer #2: Yes

4. Is the manuscript presented in an intelligible fashion and written in standard English?

Reviewer #1: No

Reviewer #2: Yes

5. Review Comments to the Author

**Reviewer #1: ** It is an interesting topic on the decision-making model in the rapid change of the environment. However, the development of the background, literature review, methods and the results are weak and need to be revisited at its entirety.

**Reviewer #2:**  1. For the proposed decision-making model, the authors should provide more details on the validation process, including sensitivity analysis or testing the model's applicability in different contexts.

2. The paper also mentions the use of fuzzy preference relationship assessment and big data analysis. The authors should describe the specific steps of these analytical methods, including data preprocessing, model training, and interpretation of results.

6. PLOS authors have the option to publish the peer review history of their article (what does this mean? ). If published, this will include your full peer review and any attached files.

**Do you want your identity to be public for this peer review?** For information about this choice, including consent withdrawal, please see our Privacy Policy .

Reviewer #1: No

Reviewer #2: No

---

## [Author Response · Author response to Decision Letter 1]

6 Jan 2025

Saddam A. Hazaea

Postdoctoral

Academic Editor in PLOS ONE

Dear Sir,

On behalf of my coauthors with excitement, I resubmit to you a revised version of our manuscript with PONE-D-24-23190, entitled “Determining Resources and Capabilities in Complex Context: A Decision-Making Model for Banks.” We appreciate the time and detail provided by each reviewer and by you and have incorporated the suggested changes into the manuscript to the best of our ability. Also, we have responded specifically to each suggestion in this document and to make the changes easier to identify where necessary, we put information about the line number.

The manuscript has certainly benefited from these insightful revision suggestions. We look forward to working with you and the reviewers to publish in PLOS ONE.

Best regards,

Authors

---

## [Decision Letter · Decision Letter 1]

5 Mar 2025

PONE-D-24-23190R1Determining  Resources and Capabilities in Complex Context: A Decision-Making Model for BanksPLOS ONE

Dear Dr. Ristyawan,

Thank you for submitting your manuscript to PLOS ONE. After careful consideration, we feel that it has merit but does not fully meet PLOS ONE’s publication criteria as it currently stands. Therefore, we invite you to submit a revised version of the manuscript that addresses the points raised during the review process.

Dear authors, please address all comments in detail.

We look forward to receiving your revised manuscript.

Kind regards,

Saddam A. Hazaea, Postdoctoral

Academic Editor

PLOS ONE

Additional Editor Comments:

Dear authors, please address all comments.

Reviewers' comments:

Reviewer's Responses to Questions

**Comments to the Author**

1. If the authors have adequately addressed your comments raised in a previous round of review and you feel that this manuscript is now acceptable for publication, you may indicate that here to bypass the “Comments to the Author” section, enter your conflict of interest statement in the “Confidential to Editor” section, and submit your "Accept" recommendation.

Reviewer #1: (No Response)

Reviewer #2: All comments have been addressed

Reviewer #3: (No Response)

2. Is the manuscript technically sound, and do the data support the conclusions?

Reviewer #1: No

Reviewer #2: Yes

Reviewer #3: Partly

3. Has the statistical analysis been performed appropriately and rigorously? 

Reviewer #1: Yes

Reviewer #2: Yes

Reviewer #3: No

4. Have the authors made all data underlying the findings in their manuscript fully available?

Reviewer #1: Yes

Reviewer #2: Yes

Reviewer #3: Yes

5. Is the manuscript presented in an intelligible fashion and written in standard English?

Reviewer #1: No

Reviewer #2: Yes

Reviewer #3: Yes

6. Review Comments to the Author

Reviewer #1: Most of the inputs in the previous review have not been addressed. The revisions were done by paraphrasing the previous paper, but not the substance of the paper as highlighted in the review. If the authors have different of opinions, then they should be highlighted in the rebuttal notes to the reviewers.

Reviewer #2: All questions have been modified specifically to enhance clarity and relevance to the study. The revisions reflect a thoughtful consideration of the feedback provided during the initial review process.

Reviewer #3: Mochammad Ridwan Ristyawan and colleagues present a conceptual model aimed at assisting bank managers in decision-making under conditions of volatility, uncertainty, complexity, and ambiguity, an issue of significant practical relevance. The integration of dynamic capabilities, socio-technical systems (STS), and big data analytics (BDA) provides a novel theoretical foundation. However, while the research framework is ambitious and methodologically rigorous in certain aspects, several critical issues must be addressed to enhance the theoretical contribution, methodological transparency, and practical applicability.

A) Major Concerns:

1 Methodological Rigor:

One key concern is the methodological rigor of the model development. In the Functional Requirement Definition (FRD) section (Table 2 in the results), the interview outlines focus heavily on the importance of addressing the problem but provide insufficient detail on the conceptual model itself. This discussion could be better positioned in the discussion section rather than in the methodological framework.

2 Model Components:

It is unclear whether all six components of the proposed conceptual framework are essential. Are all six equally critical in different decision-making contexts, or do they vary in priority depending on the situation? A more nuanced discussion on the relative importance of these components would strengthen the framework.

3 Empirical Evidence:

A significant limitation is the lack of strong empirical support for the model. In the mini-case simulation, the authors primarily describe processes, such as stating that " the BOD analyzes resource and capability requirements for the latest strategy," but do not provide concrete evidence demonstrating the model’s effectiveness. Is there empirical data showing that applying the conceptual framework leads to measurable improvements in banking operations? For instance, is there a pre- vs. post-implementation comparison, and if so, what is the effect size? Furthermore, the mini-case study is narrow in scope and lacks comparative analysis with other banks or industries. A multi-case approach would enhance the model’s generalizability.

4 Implementation Challenges:

Another critical issue is the practical implementation of the model. How can bank managers integrate this framework into real-world decision-making processes? The authors need to provide more concrete guidance or discussion on implementation strategies.

B) Minor Concerns:

Presentation of Table 5: Instead of using a table to conceptualize insights from interconnected dimensions, a network graph might offer better clarity. Specifically, the graph’s nodes could represent the relational dimensions, making the structure more intuitive.

Clarification of Theoretical Novelty: The authors should explicitly contrast the proposed model with existing frameworks and highlight why it provides a superior approach to addressing complexity and ambiguity in banking decision-making.

Addressing these issues would significantly enhance the robustness and impact of the study.

7. PLOS authors have the option to publish the peer review history of their article (what does this mean? ). If published, this will include your full peer review and any attached files.

**Do you want your identity to be public for this peer review?** For information about this choice, including consent withdrawal, please see our Privacy Policy .

Reviewer #1: No

Reviewer #2: No

Reviewer #3: No

---

## [Author Response · Author response to Decision Letter 2]

25 Mar 2025

We have taken note of the comments and made improvements based on them.

---

## [Decision Letter · Decision Letter 2]

15 Apr 2025

Determining  Resources and Capabilities in Complex Context: A Decision-Making Model for Banks

PONE-D-24-23190R2

Dear Dr. Mochammad Ridwan Ristyawan

We’re pleased to inform you that your manuscript has been judged scientifically suitable for publication and will be formally accepted for publication once it meets all outstanding technical requirements.

Kind regards,

Saddam A. Hazaea, Postdoctoral

Academic Editor

PLOS ONE

Additional Editor Comments (optional):

Congratulations, based on the review comments, your paper is now accepted.

Reviewers' comments:

Reviewer's Responses to Questions

**Comments to the Author**

1. If the authors have adequately addressed your comments raised in a previous round of review and you feel that this manuscript is now acceptable for publication, you may indicate that here to bypass the “Comments to the Author” section, enter your conflict of interest statement in the “Confidential to Editor” section, and submit your "Accept" recommendation.

Reviewer #3: All comments have been addressed

2. Is the manuscript technically sound, and do the data support the conclusions?

Reviewer #3: (No Response)

3. Has the statistical analysis been performed appropriately and rigorously? 

Reviewer #3: (No Response)

4. Have the authors made all data underlying the findings in their manuscript fully available?

Reviewer #3: (No Response)

5. Is the manuscript presented in an intelligible fashion and written in standard English?

Reviewer #3: (No Response)

6. Review Comments to the Author

Reviewer #3: (No Response)

7. PLOS authors have the option to publish the peer review history of their article (what does this mean? ). If published, this will include your full peer review and any attached files.

**Do you want your identity to be public for this peer review?** For information about this choice, including consent withdrawal, please see our Privacy Policy .

Reviewer #3: No

---

## [Editor Report · Acceptance letter]

PONE-D-24-23190R2

PLOS ONE

Dear Dr. Ristyawan,

I'm pleased to inform you that your manuscript has been deemed suitable for publication in PLOS ONE. Congratulations! Your manuscript is now being handed over to our production team.

Kind regards,

on behalf of

Dr. Saddam A. Hazaea

Academic Editor

PLOS ONE